# Study on the Influence of Regular Physical Activity on Children’s Oral Health

**DOI:** 10.3390/children10060946

**Published:** 2023-05-26

**Authors:** Paul Șerban Popa, Daniela Onișor, Aurel Nechita, Kamel Earar, Mădălina Nicoleta Matei

**Affiliations:** Research Centre in the Medical-Pharmaceutical Field, Faculty of Medicine and Pharmacy, Dunărea de Jos University of Galați, 47 Domnească Str., 800181 Galați, Romania

**Keywords:** physical activity, oral health, sports dentistry, OHI, PMA, DMF-T, football, hockey

## Abstract

The benefits of physical activities conducted systematically on the harmonious development, intellectual performance, and general health of children are unanimously accepted. This study’s aim is to determine whether differences in oral health between young athletes and children not engaged in competitive sports are present. A total of 173 children aged between 6 and 17 years, 58 hockey players, 55 football players, and 60 in the control group were divided into groups according to their activity, age, and biological sex and examined for oral hygiene and dental and periodontal health, using clinically determined indices. Statistical analysis showed significant differences between the groups, with lower (better) values for athletes, regardless of age, sex, or activity. Oral hygiene showed the most relevant differences for males aged 14 to 17, as did the index for dental health. Periodontal health, on the other hand, was significantly better for females aged 6 to 13. Based on this data, the beneficial influence of regular physical activity also has an impact on oral health. Identifying the mechanisms behind this needs to be explored in depth and may be a topic for further research.

## 1. Introduction

Sports is an elementary activity in every child’s life as it brings immediate benefits to the harmonious development and health of young people. Sports are also a powerful tool to help young people maintain their physical and mental balance.

The medical benefits of participating in competitive sports include reduced risk of cardiovascular disease and diabetes, significant contribution to weight management, improved physical condition, improved joint mobility, and better sleep quality [1,2,3,4,5]. 

Many young people who regularly engage in strenuous physical activity perform better academically [6]. Competitive sports require considerable time and energy. However, sports develop memorization, repetition, and learning skill sets that are relevant to educational activity [7]. Competitive sports involve hard work and dedication while increasing self-esteem. When a child sees their efforts appreciated, it boosts their self-confidence [8,9,10]. Playing a competitive sport is a useful way to reduce stress and ensure optimal psycho-somatic development.

Given the increasing importance attached to children’s oral health, it is appropriate to investigate the effects of physical activity on this aspect. There are several studies targeting this issue [11,12,13,14,15,16], but the approach is not uniform; there are different methodologies and sometimes contradictory results [17,18,19]. Most articles focus on changes in saliva during physical effort [20,21,22], with less emphasis on assessing oral hygiene or dental and periodontal health.

The indices used in this study to determine the level of oral hygiene (OHI-S—Oral Hygiene Index–Simplified Index) and dental and periodontal health (DMF-T—Decayed/Missing/Filled Teeth Index and PMA—Papillary-Marginal-Attached Gingival Index) are well known in the specialized literature, being widely used in epidemiological studies, providing clear and precise information on the aspects studied via a simple methodology. Elevated values of OHI-S were correlated with high values of PMA and DMF-T, both among children and adults [23,24,25,26]. Although there are numerous articles that address the issues of oral hygiene and dental and periodontal health in children or performance athletes, no relevant studies have been identified that analyze the situation of competitive athlete children [27,28,29,30,31].

Sports dentistry has undertaken steps in analyzing the impact of competitive athletic activities on oral health but is almost exclusively focused on adults. Numerous studies and systematic reviews on this topic exist, the vast majority pointing out the higher prevalence of dental caries, dental erosion, and periodontal diseases in competitive adult athletes. The main causal factors for these results were linked to modified salivary pH and flow rate, dehydration, using a mouthguard, or high sugar intake from sports supplements [32,33,34,35,36].

The main aim of this research is to determine whether there are any differences in oral health between children involved in competitive sports compared to those not engaged in such activities. An oral health analysis was conducted on two groups of children participating in competitive sports to identify potential dissimilarities, the null hypothesis (H_0_) being that there is no variability amidst them. The analysis focused on the assessment of oral hygiene as the main degradation factor for oral health, as well as on dental damage through carious processes and periodontal signs of inflammation. Football and hockey were chosen, both being popular team sports among children, sharing common characteristics such as the duration of training/matches and the type of physical effort involved (aerobic, not anaerobic). The degree of oral hygiene and dental and periodontal health were analyzed using oral health indices, which were compared both with those determined on a control group and between the two groups of athlete children. Another aim was to assess whether the conclusions available in the literature regarding oral health in adult competitive athletes are also present in much younger participants. The results disproved the null hypothesis and showed a significant positive influence of regular physical activity on children’s oral health (H_a_—alternative hypothesis), contrary to the impact of sports training in adult competitive athletes. Moreover, the data available for adults on this topic did not concur with this study’s results, competitive athlete children having better oral health than their adult counterparts.

## 2. Materials and Methods

This study involved 173 children aged between 6 and 17, who were divided into groups according to their biological sex, age, and sporting activity for more relevant analysis. A total of 58 children played hockey, 55 football, and 60 were in the control group. Applying the abovementioned criteria, 15 groups were obtained: H (all children playing hockey), F (all children playing football), C (control group), H1 (hockey players aged 6–13/24 children), F1 (football players aged 6–13/27 children), C1 (control group aged 6–13/30 children), H2 (hockey players ages 14–17/34 children), F2 (football players aged 14–17/28 children), C2 (control group aged 14–17/30 children), HM (male hockey players/34 children), FM (football male players/35 children), CM (control male children/29 children), HF (hockey female players/24 children), FF (female football players/20 children), and CF (control female children/31 children). The age of 13 was used as a distribution factor considering the physiological age of the permanent dental eruption’s completion. 

Statistical analysis of clinical examination data was performed using MedCalc software version 20.215. The normal distribution of the index values for the groups was assessed using the chi-squared test. The differences were statistically analyzed with the paired samples t-test. The minimum sample size was calculated using the following criteria: 0.2 was the value for Type I error/Alpha (Significance) and 0.1 for Type II error/Beta (1-Power).

Any child who had undertaken treatment with antibiotics, vitamins, or professional products containing fluoride (gels and varnishes) during the month prior to the examination was excluded from this study, regardless of their potential category, study groups, or control. Moreover, the presence of ulcers, tumor lesions, herpes, canker sores, or any developmental enamel defect constituted exclusion criteria from this study. Children undergoing orthodontic treatment were permitted to participate. For the athletes, the inclusion criteria in this study were the following: age between 6 and 17 years, regular/consistent practice of a sport, and a minimum duration of 1 year since the beginning of the sports activity. The exclusion criteria were the following: age outside the range of 6–17 years, lack of membership in an accredited sports organization, less than 1 year in sports activity, and presence of only deciduous teeth. 

The children involved in competitive sports were recruited from two local sports clubs: CSM Dunărea Galați (hockey) and ACS Real Union Galați (football). The control group was formed with regular pediatric patients who used the centers mentioned below for dental treatments and met the eligibility conditions regarding age, type of dentition, medical history, and health status. All the examinations were performed in the specialized outpatient clinic of the Emergency Clinical Hospital for Children, “Sf. Ioan”, Galați, and in the treatment center of the Faculty of Medicine and Pharmacy of the Dunărea de Jos University of Galați. Apart from determining the indices, data regarding age, sex, and number of training sessions per week was collected. The common demographic variables, such as race, income level, employment status, living environment, homeownership, and level of education, were gathered from the parents or legal tutors but were not included in this study. The patients were examined usually around noon, after their school program, and were instructed to refrain from brushing their teeth, eating, or drinking anything except water for at least one hour prior to the examination. Due to the intrinsically subjective nature of index determination, only two adequately trained professionals performed all clinical examinations. 

To identify and quantify pathological aspects related to the oral cavity, each subject was clinically examined, and the following indices were calculated: OHI-S (Oral Hygiene Index-Simplified), PMA gingival index, and DMF-T index.

The OHI-S index was introduced by Green and Vermillion in 1960 and modified in 1964, with the aim of assessing the presence of two etiological factors: plaque and dental calculus. This index is the sum of the Simplified Debris Index (DI-S) and the Simplified Calculus Index (CI-S), but the two scores can also be used separately. The latest version, including the criteria for classifying both debris and calculus as well as calculation examples, are provided by the WHO [37]. OHI-S (Oral Hygiene Index-Simplified) allows the separate assessment of soft and hard deposits present on the buccal surfaces of teeth 1.6, 1.1, 2.6, 3.1, and on the lingual surfaces of teeth 3.6 and 4.6, equaling six teeth on both arches, one tooth per sextant. Quantification of dental deposits can be performed visually or via staining solutions (Schiller-Pisarev, magenta, erythrosine) and by scraping the coronal surfaces examined with the dental probe, extending the examination to the contact points of the approximal coronal surfaces and including subgingival examination. 

Obtaining the value of the simplified oral hygiene index, OHI-S is accomplished by using the values of each of its components (DI-S and CI-S) divided by the number of surfaces examined and subsequent summation of the results obtained. Depending on the level of debris or dental calculus detected on the dental surfaces examined, a numerical value/score is assigned to each of them (Table 1). The value of each component (DI-S and CI-S) is calculated by summing all the values obtained from the clinical examination and dividing the result obtained by the number of examined surfaces (6).

Assessing the results obtained by applying the above formulas is conducted according to the data below (Table 2) based on the reference interval in which they fit. 

The Papillary-Marginal-Attached Gingival Index (PMA) is used to assess gingival damage. Multiple versions of this index have been proposed, but the most often used are the Schour and Massler PMA index (1947) or the modified Lobene PMA index (1964). According to Lobene (1964), the PMA gingival index is based on using inflammation as an indicator of the severity of gingival damage [38]. Gingival inflammation starts from the papilla (P), continues with the gingival margin (M), and, if it progresses, it will also involve the attached gingival tissues (A) (epithelial insertion). The PMA gingival index assessment aims to quantify the severity of inflammation of these three elements (P, M, and A) on a scale from 0 to 4 and is performed according to the following codes and criteria:0.- Normal tissue and a lack of inflammation;1.- Inflammation is present only in the interdental papilla (P), the classic signs being swelling, congestion, and discrete loss of contour;2.- Moderate inflammation of the marginal gum (M), with swelling, congestion, contour change (rounding) of the interdental papilla, and loose gingival margin, and a tendency to gingival bleeding on probing;3.- Severe inflammation of the attached gingiva (A) with obvious changes in shape and volume, with the appearance of pockets, discharge, and bleeding;4.- Very severe inflammation with ulceration and acute ulcerous-necrotic gingivitis.

This index can be used when performing the general oral examination, or it can be limited to the frontal teeth. The gingival index PMA is calculated according to the following formula:$$\mathrm{PMA}=\frac{\sum determined\;values}{number\;of\;teeth\;examined}\times 100.$$

Ideally, the value of the gingival index PMA equals 0. As the value grows, so does the gingival pathology. The PMA gingival index assessment criteria are as follows: 30% or less represents mild gingival damage; 31–60% represents medium gingival damage; 61% and more represents severe gingival damage [33]. This index’s accuracy depends on the examiner’s ability to differentiate between affected and unaffected areas, as well as assessing the lesions severity.

The DMF-T and DMF-S indices describe the prevalence of dental caries on teeth (DMF-T) and dental surfaces (DMF-S), each consisting of three components: decayed (D), missing due to caries (M), and filled (F). These indices represent the conventional method of estimating the degree of damage to dental integrity through the existence of carious processes in a population group. It is calculated by summing the three components (D, M, and F) for both permanent teeth (excluding wisdom teeth) as well as for deciduous teeth (dmf-t and dmf-s) representing the following: how many teeth/dental surfaces have carious lesions (incipient caries not included); how many teeth/dental surfaces are missing due to extraction caused by the carious process; how many teeth/dental surfaces have restorations. Each existing component has a numerical value of 1, adding up all of them represents the total index value. For children with mixed dentition, a DMF-T index for the permanent dentition and a dmf-t index for the deciduous dentition are determined. If a tooth/surface has both a filling and a carious lesion, only the carious process is considered. 

Considering the difficulties that exist in correctly assessing the proper categories of dental caries, precise definitions were required. The decayed tooth (D) component of the DMF-T/DMF-S index does not acknowledge the reversibility of early carious lesions. Due to this, early lesions cataloged as belonging to ICDAS codes 1 and 2 (first or distinct visual changes in enamel seen as a carious opacity or visible discoloration, e.g., white spot lesion and/or brown carious discoloration, not consistent with clinical appearance of sound enamel and which show no evidence of surface breakdown or underlying dentin shadowing) were evaluated with the value 0, ICDAS lesions 3–6 receiving the value 1. 

## 3. Results

For each child, the OHI-S, PMA, and DMF-T values were determined, resulting in fifteen sets of data that were statistically analyzed using MedCalc software, version 20.215, developed by MedCalc Software Ltd., Ostend, Belgium. The series was subjected to a chi-squared test to analyze the normality of the distribution. *t*-tests were conducted to establish the statistical significance of the data obtained. The relevant gathered demographic data regarding the participants are presented in the table below (Table 3). 

### 3.1. OHI-S

For the group of children playing hockey (H), the OHI-S values ranged from 0 to 3, with an arithmetic mean of 0.9797, a median of 1, and a standard deviation of 0.7405. Regarding the degree of normality of the distribution of the determined values, the chi-squared test returned 72.896, DF = 8, and *p* < 0.0001. Group F, children playing football, had OHI values between 0.3300 and 2, with an arithmetic mean of 1.4720, a median of 1.8300, and a standard deviation of 0.5005. The chi-squared test for distribution was 81.759, DF = 7, and *p* < 0.0001. The control group (C) had the minimum OHI-S 1, a maximum of 4.8300, an arithmetic mean of 2.4350, and a median of 2.6600. The distribution of values for this group respected normality, chi-squared = 35.671, DF = 9, and *p* < 0.0001.

Comparing group H with the control, the following values were obtained: a mean difference of 1.4359, a standard deviation of differences 1.2303, standard error of the mean difference of 0.1615, with *p* < 0.0001. These values reflect a statistically significant difference in favor of the hockey group, with lower OHI-S values than the control group. The same was determined for group F, football players, with the following values: a mean difference of 0.9573, a standard deviation of differences of 1.1797, a standard error of the mean difference of 0.1591, and *p* < 0.0001. It should be noted that the differences in the OHI-S index for football players compared to the control group are smaller than for children playing hockey.

Taking age and biological sex into consideration, the following results were obtained: the lowest values for *p* (<0.0001) were determined comparing the groups aged 14 to 17, both for hockey and football; biological males also had lower values for *p* than biological females, for both hockey and football. Two comparisons (F1 vs. C1 and HF vs. FF) returned irrelevant values due to not meeting the minimum sample size requirement (Table 4).

### 3.2. PMA

For the group of children playing hockey (H), the PMA index values ranged from 9.3411 to 56.0917, with an arithmetic mean of 29.6828, a median of 28.1610, and a standard deviation of 10.0893. Regarding the degree of normality of the distribution of the determined values, the chi-squared test returned 3.318, DF = 8, and *p* = 0.9129. Group F, children playing football, had PMA values ranging from 10.2631 to 61.3807, with an arithmetic mean of 37.6298, a median of 38.2096, and a standard deviation of 10.5876. The chi-squared test for distribution was 1.637, DF = 8, and *p* = 0.9902. The control group (C) had a minimum PMA value of 17.6861, a maximum of 59.2869, an arithmetic mean of 43.2632, and a median of 45.3304. The distribution of values for this group respected normality, chi-squared = 12.760, DF = 8, and *p* = 0.1204.

Comparing group H with the control, the following values were obtained: mean difference of 13.4461, standard deviation of differences of 14.9185, standard error of the mean difference of 1.9589, and *p* < 0.0001. These values reflect the existence of a statistically significant difference in favor of the group of hockey players, who have lower PMA values than the control group. The same was determined for group F, football players, with the following values: a mean difference of 5.5870, a standard deviation of differences of 12.4507, a standard error of the mean difference of 1.6788, and *p* = 0.0016. The *p*-value obtained in this comparison, although statistically relevant, does not provide the same degree of confidence as the value obtained comparing the PMA index for children playing hockey with the control group.

Considering the age and biological sex distribution, the analysis showed the following: four comparisons provided irrelevant results due to not having the minimum sample size required for statistical significance (F1 vs. C1, H2 vs. F2, FM vs. CM, and HF vs. FF); the lowest values for *p* were obtained while analyzing biological females and the children aged 6 to 13, while the highest values for *p* were determined in the analysis of biological females playing football aged 14 to 17. (Table 5)

### 3.3. DMF-T

For the group of children playing hockey (H), the DMF-T values ranged from 0 to 4 with an arithmetic mean of 1.3276, a median of 1, and a standard deviation of 1.0326. Regarding the degree of normality of the distribution of the determined values, the chi-squared test returned 106.736, DF = 9, and *p* < 0.0001. Group F, children playing football, had DMF-T values between 0 and 5, with an arithmetic mean of 2.1091, a median of 2, and a standard deviation of 1.4488. The chi-squared test for distribution was 36.315, DF = 7, and *p* < 0.0001. The control group (C) had a minimum DMF-T value of 0, a maximum of 9, an arithmetic mean of 3.4, and a median of 3. The distribution of values of this group respected normality, chi-squared = 37.584, DF = 8, and *p* < 0.0001.

Comparing group H with the control, the following values were obtained: mean difference of 1.9483, standard deviation of differences 2.2589, standard error of the mean difference of 0.2966, and *p* < 0.0001. These values reflect a statistically significant difference in favor of the hockey group, with lower DMF-T values than the control group. The same was determined for group F, football players, with the following values: a mean difference of 1.1273, a standard deviation of differences 2.5170, a standard error of the mean difference of 0.3394, and *p* = 0.0016. The *p*-value obtained in this comparison, although statistically relevant, does not provide the same degree of confidence as the value obtained comparing the DMF-T index for children playing hockey with the control group.

Analyzing the groups according to age and biological sex, the following results were obtained: the lowest values for *p* were attributed to biological men playing either sport and to children aged 14–17; higher values for *p* were determined comparing biological females and children aged 6–13. All comparisons were performed as they met the minimum sample size for statistical significance. (Table 6)

## 4. Discussion

This study’s main objective was to assess whether differences in oral health and oral hygiene exist between children involved in competitive sports and those not regularly participating in sports competitions by using oral status assessment indices. The statistical analysis of the clinically obtained results refuted the null hypothesis.

All three indices had lower (better) values for those in the performance sports groups than those in the control group. Oral hygiene, analyzed through the OHI-S index, had a mean value of 2.435 in the control group compared to 0.9797 for hockey and 1.472 for football, showing better oral hygiene for those who regularly engage in physical activities. The mean value of PMA, as a method of assessing periodontal health, was 29.6828 for hockey players and 37.6298 for football players, while the control group had a value of 43.2632. Dental health, assessed through carious lesions, was also better in children participating in sports activities compared to the control group. The DMF-T index had the following values: 1.3276 for hockey players, 2.1091 for football players, and 3.4 for the control group.

The differences observed between the two groups of children participating in sports activities, although smaller than those observed for the control group, are nevertheless statistically significant. In the case of OHI-S, the difference between the mean values is 0.4751 with *p* = 0.0007; for the PMA index, the difference between the mean values is 8.0709 with *p* = 0.0003; the DMF-T index shows a difference of 0.7636 with *p* = 0.0010.

The average age of children playing hockey was 13.67, 13,1 for football players, and 13.23 for the control group. Using age as a criterion for statistical analysis, for each index, the children involved in competitive sports aged between 14 and 17 showed better oral hygiene, periodontal health, and dental integrity than those aged 6 to 13, with a slightly higher bias on hockey players. 

A difference in biological sex was also determined, with male athletes having better index values than females, considering oral health and oral hygiene, regardless of the practiced sporting activity. 

The results obtained during this study seem to identify a positive impact of competitive sports on oral hygiene, periodontal health, and dental integrity of children, contrary to the available studies on adult athletes [32,33,34,35]. No relevant studies have been identified that analyze these impacts on competitive athlete children. Adult athletes mostly suffer from periodontal diseases [39,40], with dental caries being the second most often complaint [41,42,43]. This study’s results do not support these findings; on the contrary, periodontal health and dental integrity are better in children involved in performance sports. However, this current study does not consider the increased consumption of performance supplements and high-sugar energy drinks, a habit often found in adult athletes and incriminated as a determining factor for their poor oral hygiene.

Nonetheless, these results must be interpreted with caution, and some limitations should be borne in mind. Being a cross-sectional study, the temporal link between the outcome and the exposure cannot be accurately determined because both are examined at the same time. The differences among groups could be attributed to variations among the study subjects (different dietary or oral hygiene habits). The relatively limited number of participants, the local recruitment criteria, and the choice of competitive activities (team sports) could be potential sources of bias. This study’s methodology should be replicated on larger population groups, on a regional or even national level, including more diverse sporting activities, such as individual disciplines and not only team sports. 

However, the comprehensive analysis of the factors involved in oral health (hygiene, dental integrity, and periodontal inflammation), the homogeneous distribution of the children’s age and biological sex, and the choice of similar sports could be considered strengths of this study and should weigh favorably towards the significance of the results. 

The difference between these results and those available for adult athletes, the influence of the ambient temperature, and the differences in saliva composition should be investigated further and could be future research directions.

## Figures and Tables

**Table 1 children-10-00946-t001:** Criteria for assessing debris and dental calculus.

Score	Debris (D)	Dental Calculus (C)	Score
0	D. absent	C. absent	0
1	D. covers < ⅓ of the cervical surface examined or there is brown extrinsic dyschromia (any amount)	C. supragingival < ⅓ of the area examined	1
2	D. located between ⅓ and ⅔ of the cervical surface examined	C. located between ⅓ and ⅔ of the cervical surface examined	2
3	D. covers > ⅔ of the cervical surface examined	C. covers > ⅔ of the cervical surface examined	3

**Table 2 children-10-00946-t002:** Reference intervals for DI-S, CI-S, and OHI-S values.

Reference Interval DI-S/CI-S	Evaluation	OHI-S Index Value Interval	OHI-S Evaluation
0	excellent	0	excellent
0.1–0.6	well	0.1–1.2	well
0.7–1.8	satisfactory	1.3–3.0	satisfactory
1.9–3	unsatisfactory	3.1–6	unsatisfactory

**Table 3 children-10-00946-t003:** Demographic data.

Baseline Characteristic	Hockey Group		Football Group		Control Group	
	*n*	%	*n*	%	*n*	%
Gender						
Male	24	41.37	20	36.36	31	51.67
Female	34	58.63	35	63.64	29	48.33
Age						
6 to 13	24	41.37	27	49.09	30	50
14 to 17	34	58.63	28	50.91	30	50
Number of training sessions per week						
1–2	6	10.34	5	9.1	-	-
3–4	23	39.35	28	50.91	-	-
5 or more	29	50.01	22	39.99	-	-

**Table 4 children-10-00946-t004:** Statistical analysis for OHI-S index.Groups.

	Groups	H1/C1	F1/C1	H1/F1	H2/C2	F2/C2	H2/F2	HM/CM	FM/CM	HM/FM	HF/CF	FF/CF	HF/FF	H/C	F/C	H/F
Statistical Vales																
Mean difference		1.087	0.5248	0.623	1.7817	1.3268	1.7817	1.7979	1.2297	0.5682	1.2146	0.792	0.4155	1.4359	0.9573	0.4751
SD of differences		1.1357	0.9871	0.7636	1.0716	1.1139	1.0716	1.3418	1.1173	0.8851	0.9743	0.8775	0.8233	1.2303	1.1797	0.9809
Minimum sample size		8	35	11	4	6	4	5	7	17	6	9	23	6	11	29
Actual sample size		23	27	23	30	28	28	29	29	34	24	20	20	58	55	55
Two-tailed probability *p*		0.0001	-	0.0007	<0.0001	<0.0001	<0.0001	<0.0001	<0.0001	0.0007	<0.0001	0.0007	-	<0.0001	<0.0001	0.0007

**Table 5 children-10-00946-t005:** Statistical analysis for PMA index.

	Groups	H1/C1	F1/C1	H1/F1	H2/C2	F2/C2	H2/F2	HM/CM	FM/CM	HM/FM	HF/CF	FF/CF	HF/FF	H/C	F/C	H/F
Statistical Vales																
Mean difference		17.5451	3.6812	12.9886	12.4292	8.4492	3.174	13.9161	5.9459	9.3828	14.7337	8.6943	7.1583	13.4461	5.587	8.0709
SD of differences		11.7104	10.542	9.2924	13.4518	12.999	15.1188	14.2601	12.9556	12.0024	10.9883	12.4839	14.3291	14.9185	12.4507	15.6376
Minimum sample size		4	55	5	9	17	150	8	29	12	5	15	27	8	34	26
Actual sample size		23	27	23	30	28	28	29	23	34	24	20	20	58	55	55
Two-tailed probability *p*		<0.0001	-	<0.0001	<0.0001	0.0019	-	<0.0001	-	0.0001	<0.0001	<0.0001	-	<0.0001	0.0016	0.0003

**Table 6 children-10-00946-t006:** Statistical analysis for DMF-T index.

	Groups	H1/C1	F1/C1	H1/F1	H2/C2	F2/C2	H2/F2	HM/CM	FM/CM	HM/FM	HF/CF	FF/CF	HF/FF	H/C	F/C	H/F
Statistical Vales																
Mean difference		2.5652	1.4815	0.7826	1.2667	0.6071	1.4643	2.2069	1.3793	0.9706	1.875	1.2	0.35	1.9483	1.1273	0.7636
SD of differences		1.8787	2.3918	1.2329	2.1324	0.6153	1.5271	2.2894	2.3209	1.9146	2.309	2.4836	1.4865	2.2589	2.517	1.6326
Minimum sample size		5	18	18	20	8	8	8	20	27	11	11	9	10	34	31
Actual sample size		23	27	23	30	28	28	29	29	34	24	20	20	58	55	55
Two-tailed probability *p*		0.0021	0.0034	0.0015	0.0002	<0.0001	<0.0001	<0.0001	0.0034	0.0057	0.0006	0.0012	0.002	<0.0001	0.0016	0.001

## Data Availability

The data presented in this study are available upon request from the corresponding author. The data are not publicly available due to privacy reasons.

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
