# Peer review of "Study on the Influence of Regular Physical Activity on Children’s Oral Health"

_children, 2023, doi:10.3390/children10060946_

Round 1
Reviewer 1 Report (Previous Reviewer 2)
The authors corrected what I suggested.
Some minor typo corrections.
Author Response
Thank you very much for your review. Your input is greatly appreciated and taken into careful consideration. The typos will be rectified in the next version.
Reviewer 2 Report (New Reviewer)
abstract: there is no clear definition of the goals (which will not be in the form of questions) and there should be a breakdown of the results.
Introduction: state DMF-T and PMA the first time it is listed Abbreviation of what?
Add reference to sentence from line 42-44.
MATERIALS AND METHODS:
do children with MIH or any developmental enamel defect were exclude?
or children undergoing orthodontic treatment .?
who examined the children.one examiner or many. Maybe if there was one or two testers the results would be more reliable.
Results:
Add table with demographic data.
discussion:
add to limitations that it is a cross-sectional study. The exposure and outcome are simultaneously assessed, there is generally no evidence of a temporal relationship between exposure and outcome.
May be the difference in caries experience and periodontal disease is because of different oral hygiene and diet habits .
Author Response
Please see the attachment.

Reviewer 3 Report (New Reviewer)
Dear authors,
I had the opportunity to review your article. Apparently the article has already been revised and is being resubmitted (I did not have access to the first version).
A null hypothesis must be included at the end of the Introduction section.
The methodology is very confusing and there is no clear information about the control group. There is information about the groups of children in the test groups (where and how they were recruited, what are the inclusion and exclusion criteria, etc.) but I cannot find the same type of information for the control group.
The discussion is very poor and small. The first 5 paragraphs are essentially a repetition of the results, with no new information or interpretation/discussion. A discussion should be held in order to justify the results and compare them with similar studies already published, namely in adults, as referred by the authors as a secondary objective.
I believe some minor editing of English language is required
Round 2
Reviewer 3 Report (New Reviewer)
Thank you for your answers and for the update of the new version of the manuscript.
Minor editing of English language is required.
Author Response
Thank you very much for your insights and valuable guidance and comments. Final proofing of the manuscript will be conducted prior to submission.
This manuscript is a resubmission of an earlier submission. The following is a list of the peer review reports and author responses from that submission.
Round 1
Reviewer 1 Report
Thank you very much for your article with this interesting research question. Unfortunately, with the current available information, I do not consider the presented manuscirpt appropriate and your study do not seem to be able to answer your research question.
Please consider the STROBE guidelines for reporting your study.
Introduction: Evidence on oral health in adult competitive athletes should be included (higher prevalence of oral inflammation and caries, please see the systematic reviews in this topic).
Material and Methods: You could shorten the description of the indices if you present the reference with the description. Please add a reference for the OHI-S. How did you recruited the participants (in sports clubs? local/national?). Under which conditions did you perform the examinations? What was your treshold between incipient caries and decayed tooth. Please add you definitions. Which further information of the children did you collect (age, sex, hours of sports, ...?).
Results: You should present more Results about the participants. Especially information such as age, sex, training hours per week, .. should be presented per groups as they could be the reason for the group differences. Please add prepared tables with the results in the main manuscript. Screenshots of a statistic program are no proper presentation of your results.
Discussion: Here the results should be discussed ("Authors should discuss the results and how they can be interpreted in perspective of previous studies and of the working hypotheses. The findings and their implications should be discussed in the broadest context possible and limitations of the work highlighted. Future research directions may also be mentioned") . Please consider strengths and limitations of the study. The influence of group differences in age, sex, socioeconomics, ... should be discussed here as potential source of bias. You should compare your results with already published data of athletes as well as of children in general.
Conclusion: With the currently included information your study is not able to judge about any influence of sports on oral health.
Reviewer 2 Report
Dear Authors,
It is an interesting topic. However, some issues need to be changed before considering it for publication:
1. Introduction
Please find previous studies that used OHI, PMA, and DMFT values to assess oral hygiene, dental and periodontal health and describe results.
Lines 40- 43. You write: „The degree of oral hygiene, dental and periodontal health were analyzed, using oral health indices, which were compared with those determined on a control group and between the two groups of children who participate in sports activities. The results show a significant positive influence of regular physical activity on children's oral health.”
That is not part of the introduction. Please, describe your aim better: which sports? and what oral health analysis?
2. Materials and Methods
Line 53 Not “temporary” but “deciduous” (please change it in the whole text).
How did you calculate the sample size?
You divided children into three groups, but oral health status, dental and periodontal are very different between 6 year- old children and 17-year-old children. It is better to have more groups according to childrens years, for example, 6-9 years old, 10-13 years old, and 14-17 years old. In that case, it can happen that you may not have enough statistical samples for each group to have adequate statistical analyses, and you may need more participants in the study.
3. Results
Please make Tables for OHI-S, PMA, and DMF with results. You may not use the Figures that you sent in supplementary materials.
4. Discussion
Please find previous studies that used OHI, 11 PMA, and DMFT values to assess oral hygiene, dental and periodontal health and compare it, and discuss with your results.
What are the limitations of your study and strengths of your study?